# Four New Species of Dictyostelids from Soil Systems in Northern Thailand

**DOI:** 10.3390/jof8060593

**Published:** 2022-05-31

**Authors:** James C. Cavender, Eduardo M. Vadell, Allison L. Perrigo, John C. Landolt, Steven L. Stephenson, Pu Liu

**Affiliations:** 1Engineering Research Center of Edible and Medicinal Fungi, Ministry of Education, Jilin Agricultural University, Changchun 130118, China; 2Departmental of Environmental and Plant Biology, Ohio University, Athens, OH 45701, USA; cavender@ohio.edu; 3Museo de Historia Natural R.S.V.—Microbiology, Suipacha 6612 Pcia. de Buenos Aires, Viamonte 1716 6p, Buenos Aires City, Argentina; eduardo.vadell@gmail.com; 4Gothenburg Global Biodiversity Centre, Box 461, 405 30 Gothenburg, Sweden; allison.l.perrigo@gmail.com; 5Department of Biology, Shepherd University, Shepherdstown, WV 25443, USA; landolt@shepherd.edu; 6Department of Biological Sciences, University of Arkansas, Fayetteville, AR 72701, USA; slsteph@uark.edu

**Keywords:** *Cavenderia*, cellular slime molds, species concept, taxonomy, biodiversity

## Abstract

Dictyostelid cellular slime molds (dictyostelids) are ubiquitous microorganisms found in the uppermost layers of most soils. Reports on the species diversity of dictyostelids in Southeast Asia, particularly Thailand, are few in number. A survey for dictyostelids performed in northern Thailand in 2008 recovered 15 distinctive forms, including several common species and a number of forms morphologically different from anything already described. Five of the latter were formally described as new to science in a previous paper. An additional five isolates appeared to be morphologically distinct, and this was supported by DNA sequence data and phylogenetic analysis. These isolates representing four species are described herein as species new to science. Detailed descriptions and illustrations of these new species are provided.

## 1. Introduction

Dictyostelid cellular slime molds (dictyostelids) are a ubiquitous component of soils, where they feed upon bacteria and other microbes and thus play a major role in maintaining the balance between these organisms in the soil microhabitat [1,2]. Dictyostelids are amoebozoans, a distinct branch of eukaryotes, separate from plants, fungi, and animals. During the past 20 years, as a result of ongoing surveys in areas of the world where dictyostelids remain an understudied group, the number of species has essentially doubled. Molecular studies performed over the past decade have revised the traditional system of classification used for dictyostelids [3], and a SSU-based phylogeny has established that there are four major clades [4].

In 2008, a total of 40 samples for isolation of dictyostelids were collected from four localities in northern Thailand to obtain data on the occurrence and distribution of dictyostelids in this region of the world. One of the localities sampled was located within a tropical cloud forest in Doi Inthanon National Park, Chom Thong District, Chiang Mai Province (18°35′32″ N, 98°29′12″ E). In addition to several common species of dictyostelids, nine isolates that could not be assigned to any described species were also recovered from the sampling. All of these isolates were subjected to a detailed morphological study of subcultures in addition to DNA sequence analyses, and five were described as new to science in a previous publication [5]. Five additional isolates, all obtained from samples collected from the tropical cloud forest (elevation 2500 m) in Doi Inthanon National Park, were also considered be morphologically distinct, albeit by only minor differences, and required additional investigation.

## 2. Materials and Methods

### 2.1. Sampling

Samples of soil/litter were collected from a tropical cloud forest in northern Thailand in January 2008. These samples, each approximately 30–50 g, were placed in sterile whirl-pack plastic bags, returned to the laboratory, and processed as soon as possible, as recommended by Cavender and Raper [6].

### 2.2. Isolation and Cultivation

The samples were processed using the methods described by Cavender and Raper [6]. Each sample was weighed, and enough sterile distilled water was added to obtain an initial soil/water dilution of 1:10. This mixture was shaken to disperse the material and to suspend the cells of the dictyostelids present. A 5.0 mL volume of this initial dilution was added to 7.5 mL of sterile, distilled water to create a 1:25 dilution of sample material. Aliquots (each 0.5 mL) of this suspension were added to each of two or three 95 × 15 mm Petri dishes prepared with hay (leached and dried, mostly Poa sp.) infusion agar [2]. This produced a final dilution of 0.02 g of soil per plate. Approximately 0.4 mL of a heavy suspension of 12–24 h *E**scherichia coli* was added to each culture plate, and plates were incubated under diffused light at 20–25 °C. Each plate was examined at least once daily for several days following the appearance of initial aggregations, and the location of each aggregate clone was marked. Aggregations, pseudoplasmodia, and sorocarps appeared in the plates over a period that ranged from 2 d to 3 w. Isolates of interest were subcultured from spores on low-nutrient agar with *E. coli*. Spores were also conserved in tubes of silica gel granules at 4 °C, as described by Raper [2]. Isolates considered to be of interest were studied in more detail, which involved an initial characterization of morphological features and obtaining a first set of images of morphological structures. Later, subcultures of these isolates were sent to Vadell and Liu. All isolates considered in the present study were deposited in the Dicty Stock Center at Northwestern University in the United States and then at Jilin Agricultural University (HMJAU) in China.

### 2.3. Morphological Observations

The characteristic stages in the life cycle, including cell aggregation and the formation of pseudoplasmodia and sorocarps, were photographed in the Liu laboratory under a dissecting microscope (Axio Zoom V16, Carl Zeiss Microscopy GmbH, Göttingen, Germany) with a 1.5× objective and a 10× ocular. Slides with sorocarps were prepared with water as the mounting medium. Features of spores, sorophores, and sorocarps were observed and measured on the slides by using a light microscope (Axio Imager A2, Carl Zeiss Microscopy GmbH, Göttingen, Germany), with a 10× ocular and 10, 40, and 100× (oil) objectives. Photographs were taken with a Axiocam 506 color microscope camera (Carl Zeiss Microscopy GmbH, Göttingen, Germany).

Detailed examinations of the development and overall morphology of subcultured clones of the five isolates from Doi Inthanon National Park were performed. These isolates were observed at magnifications from 50 to 200×, either in hydric conditions or in xeric media conditions to observe the effects of dehydration and conservation of the hydric contents of the sori over time, migration, stolon formation, and sorogen development. Observations of early aggregations and fruiting bodies were made after 2–30 d incubation under diffuse illumination at 18–26 °C. Optimal temperatures for growth and fruiting body formation, when determined, were measured in an incubator at 18, 20, 24, and 26 (±0.5 °C), under low-diffuse illumination. All major stages of development and behavior (in young and old cultures) of each isolate were observed, described, measured, and drawn by hand in India ink, carefully noting the patterns and shapes of early and late sorogens and behavior [7,8]. The morphological behavior was compared and contrasted with independent observations. Structures were studied through the top, side, and bottom of the Petri dish, making use of variations in light intensity to discriminate the various special perspectives and differences (e.g., basal zones, type and arrangement of cells or the accumulations of slime supporting the base of the sporophore, presence of granulated slime and cushions, relative degree of hydric conditions, and mound shapes) [8], using magnifications provided by the 4 and 12× lenses of a dissecting microscope along with observations made with a compound phase contrast microscope. The general criteria used for the features observed were based on Raper [2], and the drawings of the four species described herein are presented.

### 2.4. DNA Isolation, PCR Amplification and Sequencing

The spores of all five isolates being studied were collected with a sterile tip and mixed with the lysis buffer of the MiniBEST Universal Genomic DNA Extraction Kit Ver.5.0 (Takara Bio Inc., Kusatsu, Japan) following the manufacturer’s protocol. The genomic DNA solution was used directly for the small subunit (SSU) PCR amplification using the primers 18SF–A (AACCTGGTTGATCCTGCCAG) and 18SR–B (TGATCCTTCTGCAGGTTCAC) [9] along with D542F (ACAATTGGAGGGCAAGTCTG3) and D1340R (TCGAGGTCTCGTCCGTTATC) [4]. PCR products were sent to Sangon Biotech Co., Ltd. (Shanghai, China) for sequencing. Sequences obtained were deposited in the GenBank database. The isolates and the NCBI GenBank accession numbers of SSU DNA sequences considered in present study are listed in Table 1.

### 2.5. Phylogenetic Analysis

The five newly generated sequences were checked and then submitted to GenBank, as noted above. The SSU sequences were aligned and compared using the program ClustalW Multiple alignment version 2.1 (Institut Pasteur, Paris, France) [10] and then manually adjusted in BioEdit version 7.0.9.0 (Manchester, UK) [11]. Maximum likelihood (ML) analyses were performed using IQ-TREE v.1.6.12 (Insitut Pasteur, Paris, France) [12] with 1000 replicates of ultrafast-likelihood bootstrapping to obtain node support values by the “-bb 1000” option, and further optimized using a hill-climbing nearest-neighbor interchange (NNI) by the “-bnni” option [13]. The “-nt AUTO” option was used to automatically determine the best number of cores given the current data. In the ML analyses of SSU sequences, TVMe+R5 (IQTree, Vienna, Austria) model was chosen as the best-fit model according to BIC by IQ-TREE with the “-m TVMe+R5” option.

### 2.6. Data Availability

Sequence data are available in GenBank (www.ncbi.nlm.nih.gov/genbank/, accessed on 14 February 2022, Accession Numbers OM677255, OM677256, OM677257, OM677258, and OM677259). The nomenclature of the new species in the present study is available in MycoBank (www.mycobank.org, accessed on 13 February 2022 for MB842981, MB842982 and MB842984, accessed on 14 February 2022 for MB842989). Sequence alignment was uploaded as Appendix A. 

## 3. Results

Five isolates representing four new species [*Cavenderia helicoidea* (Figure 1 and Figure 2), *C. parvibrachiata* (Figure 3 and Figure 4), *C. protumula* (Figure 5 and Figure 6) and *C. ungulata* (Figure 7 and Figure 8)] of dictyostelids were recovered from samples collected from a tropical cloud forest in northern Thailand. Morphological characteristics and phylogenetic studies of the SSU sequences both support the taxonomic placement in *Cavenderia* of all four of the species (Table 2, Figure 9).

### Taxonomy and Phylogeny

***Cavenderia helicoidea*** Cavender, P. Liu, Vadell, A. L. Perrigo, J. C. Landolt & S. L. Stephenson, sp. Nov.

MycoBank accession number: MB842981; GenBank Accession Number: OM677255 (SSU), Figure 1 and Figure 2.

Etymology: The specific epithet is derived from the ancient Greek word *helix*, meaning twist or turn, and referring to the helical ascending movement during the development of the sorogens and sorocarps.

The culture examined was from: Thailand, Chiang Mai Province, Chom Thong District, Doi Inthanon National Park, 18°35′32″ N 98°29′12″ E, tropical cloud forest, isolated from a sample collected by Stephenson SL in January 2008, Landolt TH19B (Holotype) deposited in dictyostelid collection at Jilin Agricultural University (HMJAU), ex-Landolt TH19B deposited in the Dicty Stock Center at Northwestern University (No. DBS0350782).

Sorocarps were mostly prone, decumbent, helical, solitary, or clustered to gregarious, 0.23–2.85 mm long, zero to five branches (2C). Sorophores were hyaline to white or slightly yellow, with one to several tiers of cells, sometimes with a few branches. Branches 0.4–0.7 mm long. Tips acuminate or piliform with one or two tiers of cells, 4.8–5.6 μm wide (2D). Bases were clavate to round, consisting of one to six cells, with one or more terminal cells protruding, 13.3–23.3 μm wide (2E). Sori were hyaline-white to cream, globose, with a 39–241 μm diam. Spores of fallen sori germinated immediately. Spores were elliptical-oblong, 5.6–10.5 × 2.9–4.4 μm, with irregular consolidated polar granules (PG), occasionally slightly subpolar, germinating rapidly (2F). Typical myxamoeba (2G). Aggregations radiated or took the shape of irregular mounds, as psedoplasmodia with stalks (2A). Sorogens were in tight clusters. Early and late sorogens may migrate, leaving traces of slime and sections of immature sorophores (2B).

Comments: This species appears morphologically most similar to *C. protodigitata,* but the latter is smaller in length. Both share the fast fading of the yellow pigment very early during morphogenesis. This fading of the yellow pigment is a common feature among the smaller members of the Thailand species in group 1 (*sensu* Sheikh et al. [3]) that have been studied, as a relict feature tending to disappear. This isolate is also larger than *C. ungulata* and *C. aureostabil**is* [5], having similarly elongated, thin fruiting bodies also common in both species, and its sorocarps are more regular, sigmoid, and broken, and the bases are more variable. Spores are relatively large, and do not germinate immediately.

***Cavenderia parvibrachiata*** Cavender, P. Liu, Vadell, A. L. Perrigo, J. C. Landolt & S. L. Stephenson, sp. nov.

MycoBank accession number: MB842982; GenBank Accession Numbers: OM677256 (SSU), OM677257 (SSU), Figure 3 and Figure 4.

Etymology: The specific epithet *parvibrachiata* refers to the poor and small production of branches and is derived from the Latin words *parvus* (small, low, poor) and *brachium* (branch).

The culture examined was from: Thailand, Chom Thong District, Chiang Mai Province, Doi Inthanon National Park, 18°35′32″ N 98°29′12″ E, tropical cloud forest, isolated from a sample collected by Stephenson SL in January 2008, Landolt TH20C (Holotype) deposited in the dictyostelid collection at Jilin Agricultural University (HMJAU), ex–Landolt TH20C deposited in the Dicty Stock Center at Northwestern University (No. DBS0350791). Landolt 2019TH20C represents the same taxon and was isolated from the same sample that yielded Landlt TH20C.

Sorocarps were solitary or clustered to gregarious, erect to prostrate, 0.3–6.0 mm, mostly unbranched but sometimes irregularly branched, yellowish (4C). Branches were 0.4–1.2 mm. Sorophores were tenuous, slender, first straight and then curved, with one or two tiers of cells. Tips were obtuse, capitate, or clavate with one or two tiers of cells, 3.2–9.7 μm wide (4E). Bases were round, clavate, or curved hook-shaped with one to several tiers of cells, always surrounded by a layer of dense granular slime, 8.4–27.6 μm wide (4D). Sori pearls were white to slightly yellow, subglobose to globose, varying considerably in diam. (43–252 μm). Spores were elliptical-oblong, sticky, 5.4–9.7 × 2.6–4.8 μm, with pronounced consolidated subpolar to polar PG (4F). Aggregation radiated, and pseudoplasmodia was yellowish with stalks (4A). Sorogens were clustered and sometimes bifurcated (4B). Myxamobae were small at first, well-dispersed, and active, soon rounding up, with many vacuoles and a few dark granules (4G).

Comments: *Cavenderia parvibrachiata* belongs to dictyostelid group 1 in an SSU ribosomal DNA (rDNA) phylogeny (Figure 9). It forms a clade together with *C. bhumiboliana* and *C. protodigitata* and several other species. However, it differs morphologically from *C. bhumiboliana* in the width of the sorophores, branches, and spore size. *Cavenderia bhumiboliana* has tips that are 15–25 μm wide at the apex and 30–45 μm wide at the base. In *C. parvibrachiata*, the tips are 3.2–10.5 μm wide at the apex and 8.4–27.6 μm wide at the base. *Cavenderia bhumiboliana* has one to four branches whereas *C. parvibrachiata* has irregular branches. *C**. parvibrachiata* differs morphologically from *C. protodigitata* with respect to sorophore tips, branches, sorocarps, and spores. *C**. protodigitata* has piliform filaments or irregularly capitate tips. *C**. protodigitata* is also unbranched or has secondary branches.

***Cavenderia protumula*** Cavender, P. Liu, Vadell, A. L. Perrigo, J. C. Landolt & S. L. Stephenson, sp. nov.

MycoBank accession number: MB842989; GenBank Accession Number. OM677258 (SSU), Figure 5 and Figure 6.

Etymology: The specific epithet *protumulus* refers to the production of elevated small mounds (*tumulus* in Latin) that surround the bases of sorocarps.

The culture examined was from: Thailand, Chom Thong District, Chiang Mai Province, Doi Inthanon National Park, 18°35′32″ N 98°29′12″ E, tropical cloud forest, isolated from a sample collected by Stephenson SL in January 2008, Landolt TH20A (Holotype) deposited in dictyostelid collection at Jilin Agricultural University (HMJAU), ex-Landolt TH20A was deposited in the Dicty Stock Center at Northwestern University (No. DBS0350786).

Sorocarps were solitary to gregarious, erect or inclined, thin, 0.5–5.6 mm, had zero to two branches, mostly in tight clusters, sometimes colligated and coremiform, and stoloniferous (6A). Smaller solitary unbranched sorocarps, varying greatly in length, occur close to or at the base of the larger ones (6C). Sorophores had one to several tiers of cells. Branches were delicate, sigmoid, and well separated, consisting of one tier of a few cells (one to six but mostly four), arising at a right angle, 0.4–0.6 mm. Tips were obtuse and capitate, with one or two tiers of cells, 5.2–11.9 μm wide (6F). Bases were variable, regular or irregular, clavate or round with several tiers of cells, and 27.5–39.2 μm wide (6E). Sori were white, globose, 42–419 μm wide, sometimes with protruding larger cells, immersed in a bell-shaped dense granular matrix of slime. Spores were elliptical, 5.2–8.9 × 2.5–3.9 μm, with PG (6G). Typical myxamoeba (6H). Aggregation radiated and pseudoplasmodia migrated with stalks (6B). Sorogens in tight clusters (6D).

Comments: This species belongs to dictyostelid group 1 in an SSU rDNA phylogeny (Figure 9). It forms a clade together with *C. bhumiboliana* and *C. protodigitata* and several other species. However, it differs morphologically from *C. bhumiboliana* in the tips of the sorocarps, the bases of sorophores, and the sori. Sorocarps of *C. bhumiboliana* are prostrate, but *C. ungulata* has erect sorocarps. *C. bhumiboliana* has one tier of cells on the sorophores, whereas *C. protumula* has one to several tiers of cells. *Cavenderia bhumiboliana* has tips that are 15–25 μm wide, and *C. protumula* has tips that are 5.2–11.9 μm wide. *C. bhumiboliana* has bases that are 30–45 μm wide, while in *C. protumula,* they are 27.5–39.2 μm wide. *C. bhumiboliana* has sori 20–180 μm in diam, but those of *C. protumula* are 42–419 μm in diam. *Cavenderia protumula* also differs morphologically from *C. protodigitata* with respect to features of the sorocarps, the bases of the branches of the sorophores branches, and the sori. *Cavenderia protodigitata* has prostrate sorocarps, one tier of cells in the sorophores, is unbranched or has secondary branches, and sori are 50–150 μm in diam.

***Cavenderia ungulata*** Cavender, P. Liu, Vadell, A. L. Perrigo, J. C. Landolt & S. L. Stephenson, sp. nov.

MycoBank accession number: MB842984; GenBank Accession Number. OM677259 (SSU), Figure 7 and Figure 8.

Etymology: The specific epithet *ungulata* is derived from the Latin word *unguis*, meaning claw or finger claw, referring to the cat’s claw shape of the sharp terminal cells of the bases of the sorocarps.

The culture examined was from: Thailand, Chom Thong District, Chiang Mai Province, Doi Inthanon National Park, 18°35′32″ N 98°29′12″ E, tropical cloud forest, isolated from a sample collected by Stephenson SL in January 2008, Landolt TH18B (Holotype) deposited in dictyostelid collection at Jilin Agricultural University (HMJAU), ex–Landolt TH18B deposited in the Dicty Stock Center at Northwestern University (No. DBS0350791).

Sorocarps were erect to prone, solitary, clustered to gregarious, sinuous, 0.4–5.0 mm, slightly phototrophic, and mostly unbranched but sometimes with one to five branches (8D). Sorophores were extremely delicate, tenuous, uneven, white with one or two tiers of cells, slender and regular when young or commonly very irregular, consisting of one tier of cells (8C). Smaller solitary sorocarps were unbranched and varying considerably in height, sometimes surround the bases of larger sorophores (8B). Branches were 0.5–1.0 mm long. Tips were variable, obtuse to acuminate or clavate, with one to two tiers of cells (mostly one), 6.1–10.0 μm wide (8E). Bases were clavate with one or two tiers of cells, 8.9–17.2 μm wide (8F). Sori were pale white, globose, and 29–182 μm wide. Spores were elliptical-oblong, 5.0–9.8 × 2.3–4.6 μm, with prominent, large, consolidated PG, sticky, and often surrounded by a clear narrow halo (8G). Aggregations radiated, consisting of irregular mounds with short irregular streams (8A). Pseudoplasmodia migrated. Myxamoebae had many median vacuoles (8H).

Comments: *Cavenderia ungulata* is separated from any other known species of dictyostelids by its very irregular sorophores with claw-like clavate bases, sticky regular spores with large regular PG (2 µm), with halos and dispersed small vacuoles in the spore body, a homogeneous content, continuous development and growth, and migration of early sorogens. This species adapts well to different conditions, and aggregations are either those with small irregular radiate streams or tenuous aggregations that develop dendroid ample streams partitioning into small, thick, round, or slightly elongated separate pseudoplasmodia. Some features resemble those of *C. aureostipes*, although this species is very small, crowded, and delicate. This species is much smaller than *C. aureostabil**is* [5] and even smaller than *C. helicoidea* with a different spore morphology and aggregation.

## 4. Discussion

Historically, species of dictyostelids have been described solely on the basis of morphological characters [2] and only recently has it been possible to complement such morphological information with DNA sequence analysis. In general, DNA sequence data have supported morphology-based species determination of dictyostelids. However, DNA sequence information recently has led to an extensive revision of the higher levels of classification within the dictyostelids [3].

In little more than 50 years, the number of known species of dictyostelids has increased from about 40 to more than 160. Surveys for these organisms performed in regions of the world where previous investigations were either limited or completely lacking [14,15,16,17] largely account for this dramatic increase. However, more intensive studies in areas where previous records of dictyostelids already existed have yielded unexpected results. For example, Cavender et al. [18] reported 10 new species of dictyostelids from the Great Smoky Mountains National Park in the eastern United States.

In an earlier survey of dictyostelids in Southeast Asia performed in 1970 by Cavender, samples were collected from three localities in Thailand [19]. These were Kao Yai National Park in southern Thailand along with collecting sites near Chiang Dao and Chiang Mai in northern Thailand. All three sites were characterized by tropical semi-deciduous forests, and only Chiang Dao is located at an elevation comparable to that of Doi Inthanon, where the samples considered herein were collected. The data from this survey were summarized in Cavender [19]. A total of nine species were isolated, including a new group 1 species (*Cavenderia bifurcata*) from Chiang Dao.

The sequences obtained for the four species described in the present paper appear to indicate that they are closely related and probably evolved in situ. A similar situation was reported by Cavender et al. [18] for several morphologically similar species collected in a single small collecting site located at higher elevations the Great Smoky Mountains National Park in the United States. As a general observation, the species composition and diversity of the assemblage of dictyostelids in northern Thailand are relatively similar to what has been reported in previous studies of both the dictyostelids of Southeast Asia [19] and the American tropics [15]. In both regions, those species that appear to be endemic are rare. The four new and apparently rare species isolated from the tropical cloud forest at Doi Inthanon appear to be surviving as organisms adapted to a cool environment characterized by high levels of organic matter.

The major contribution of the present study is that it adds four new species of dictyostelids to the known biodiversity of these organisms in Thailand, all of Southeast Asia, and the world as a whole. The sampling effort involved certainly was not adequate to characterize the entire assemblage of dictyostelids present in the general study areas investigated. As such, it represents a potential starting point for additional future studies.

## Figures and Tables

**Figure 1 jof-08-00593-f001:**
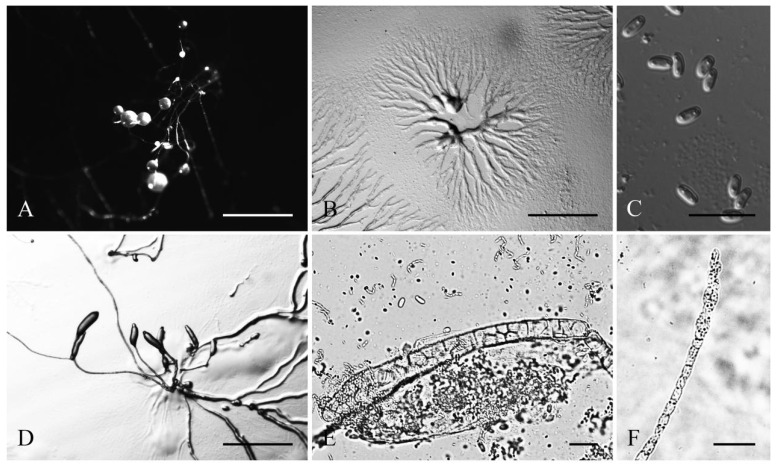
Morphological features of *Cavenderia helicoidea* (TH19B). (**A**), Sorocarps. (**B**), Aggregations. (**C**), Spores. (**D**), Clustered pseudoplasmodia. (**E**), Sorophore base. (**F**), Sorophore tip. Bars: (**A**,**B**) = 1 mm; (**C**) = 20 μm; (**D**) = 1 mm; (**E**,**F**) = 20 μm.

**Figure 2 jof-08-00593-f002:**
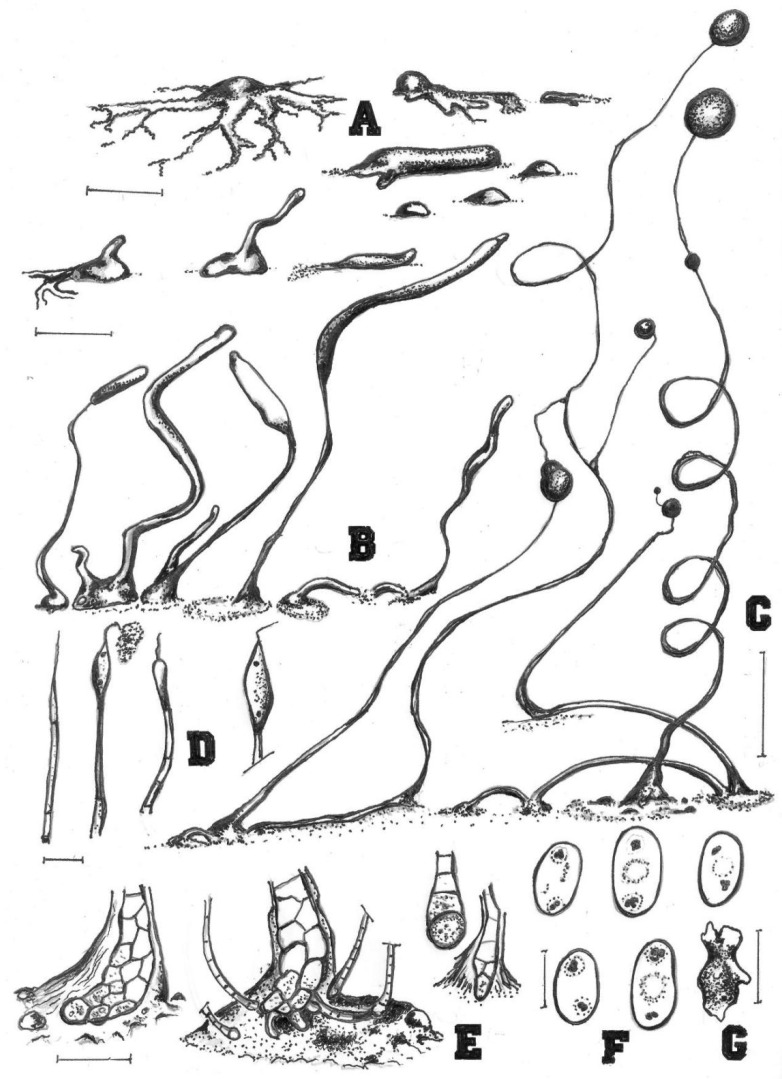
Morphological features of *Cavenderia helicoidea* (TH19B). (**A**), Medium-sized streamed aggregation, with edges (left) and the most common type consisting of small mounds, some with short blocky streams (right). (**B**), Early central sorogen (above, left); these settle without streams (above, center) and migrate with short stalk formation; small mounds, unstreamed (above, right); Group of solitary (below left and center) and tightly clustered sorogens (below center); the stoloniferous migrating habit of a late sorogen (below, right). (**C**), Clustered sorocarps, one with a prostrate lower stalk (left); two solitary sorocarps, one with a typical stoloniferous habit and the other with its frequent helical architecture; two sori are tangled, one small example refruits, and small mounds are present within the halo at bases (right). (**D**), Four simple tips, mostly piliform, and one ampule-shaped, a mass of dense slime attached to a filament. (**E**), Curved clavate base with flexuous small cells, except the terminal one (left); a curved base with small satellite sorocarps and some digitate cells, within a dense matrix of slime (center, aged culture); a one-celled round base and a clavate base, both larger and flexuose (right); all flexuose. (**F**), Elliptical slightly larger spores with prominent irregular PG. (**G**). Myxamoeba. Bars: (**A**) = 300 µm; (**B**) = 200 µm; (**C**) = 0.5 mm; (**D**) = 10 µm; (**E**) = 20 µm; (**F**) = 10 µm; (**G**) = 5 µm.

**Figure 3 jof-08-00593-f003:**
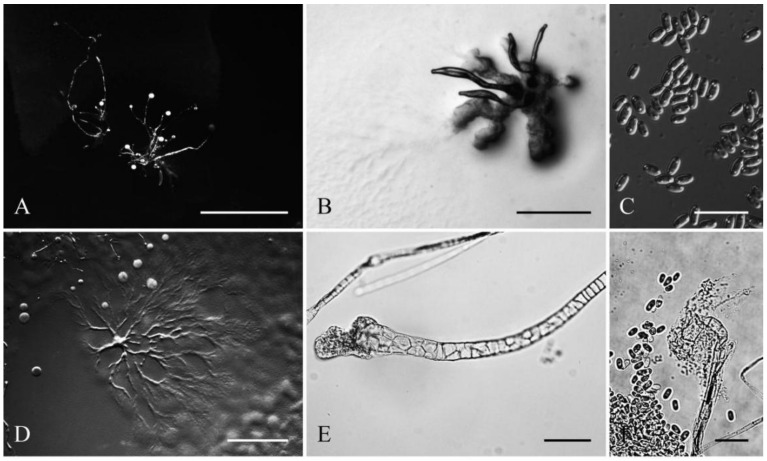
Morphological features of *Cavenderia parvibrachiata* (TH20C). (**A**), Sorocarps. (**B**), Clustered pseudoplasmodia. (**C**), Spores. (**D**), Aggregations. (**E**), Sorophore base. (**F**), Sorophore tip. Bars: (**A**) = 2 mm; (**B**) = 500 μm; (**C**) = 20 μm; (**D**) = 2 mm; (**E**) = 40 μm; (**F**) = 20 μm.

**Figure 4 jof-08-00593-f004:**
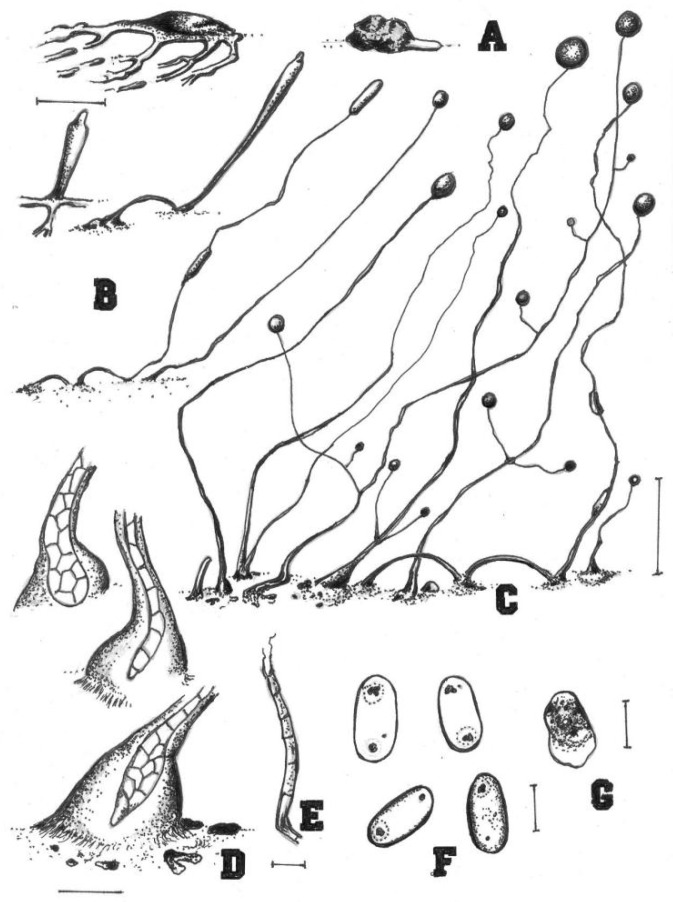
Morphological features of *Cavenderia parvibrachiata* (TH20C). (**A**), Small shortly radiated aggregation with partite streams (left); very short blocky-streamed mound-like aggregation (right). (**B**), Early central sorogen with short streams at base (above left), a stoloniferous early late elongated sorogen (above, right); stoloniferous habit of migrating late sorogen with an ascending pseudoplasmodial mass, stalk broken from early stages (below, left); solitary unbranched early mature slender small sorocarp (below, right). (**C**), Crowded group of sorocarps; tight clustered unbranched sorocarps within numerous smaller curved-broken fruiting bodies and sorogens (left); lower branched sorocarps, branches are short, sometimes coincident at one point (center); a stoloniferous sorocarp with a more evident broken sorophore and its companion small fruiting body (right). (**D**), Round base (left, above) clavate one-celled sorophore base (center); acutely clavate regular base within a hyaline mass of slime, sheath is not profuse and there are small masses of slime at base (below). (**E**), Simple tip frequently with a small cell with two piliform ends, flexuous. (**F**), Short small elliptical spores with irregular consolidated PG, one with halo. (**G**), Myxamoeba. Bars: (**A**,**B**) = 200 µm; (**C**) = 0.5 mm; (**D**) = 20 µm; (**E**) = 10 µm; (**F**) = 5 µm; (**G**) = 10 µm.

**Figure 5 jof-08-00593-f005:**
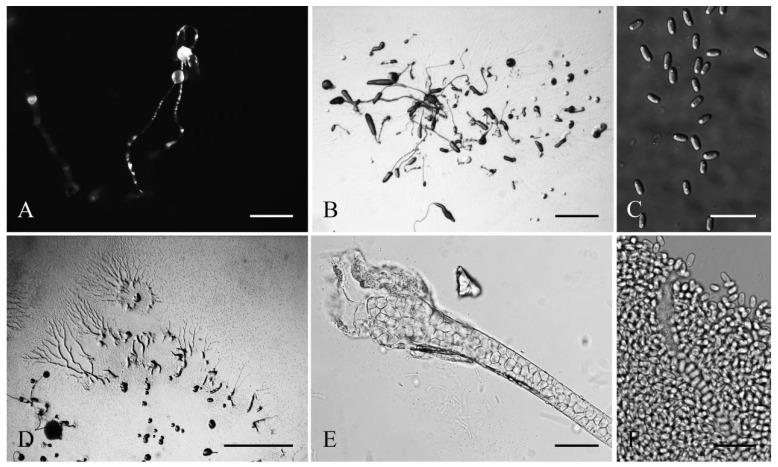
Morphological features of *Cavenderia protumula* (TH20A). (**A**), Sorocarps. (**B**), Clustered pseudoplasmodia. (**C**), Spores. (**D**), Aggregations. (**E**), Sorophore base. (**F**), Sorophore tip. Bars: (**A**) = 200 μm; (**B**) = 1 mm; (**C**) = 20 μm; (**D**) = 2 mm; (**E**) = 40 μm; (**F**) = 20 μm.

**Figure 6 jof-08-00593-f006:**
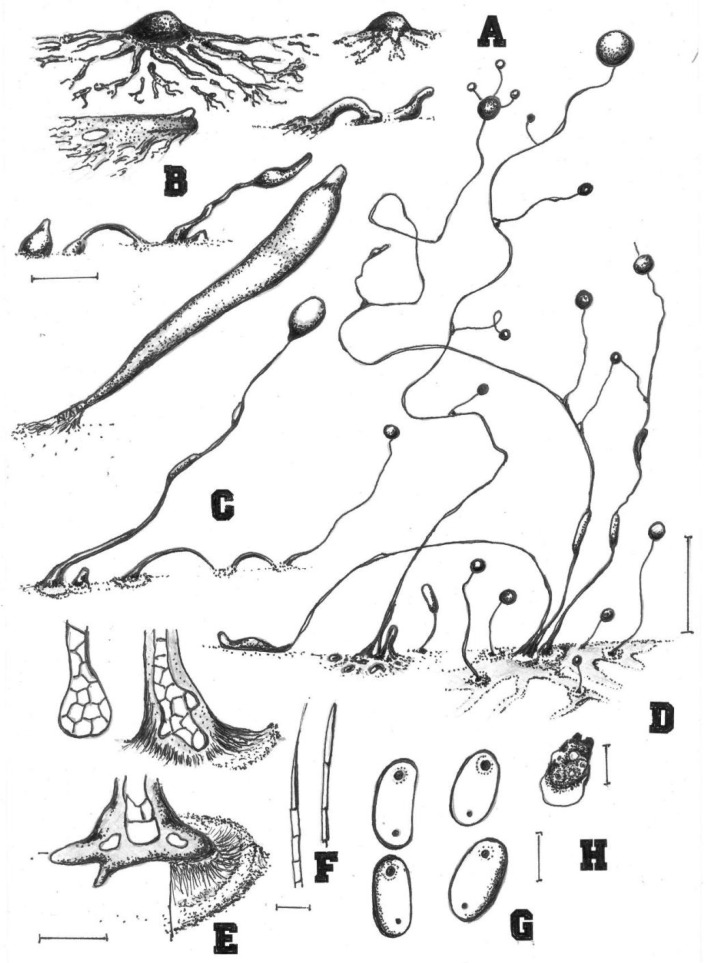
Morphological features of *Cavenderia protumula* (TH20A). (**A**), Radiate aggregation with partite streams (left); a small aggregation with short blocky streams (right). (**B**), Migrating solitary pseudoplasmodium with flat anastomosed streams (above, left); a stoloniferous migrating early sorogen (above, right); progression of an early-late solitary migrating sorogen (below, left); a single enlarged early late migrating sorogen with stalk formation at its basal portion (out of scale, below left). (**C**), Late solitary sorogen with masses of ascending pseudoplasmodia and a companion small pseudoplasmodium mass close to the base (left); stoloniferous habit of a solitary sorocarp, basal halo present (right). (**D**), Solitary sigmoid low-branched mature sorocarp with its satellite early sorogen at base, surrounded by small pseudoplasmodial mases (left); clustered sorocarps with small satellite unbranched sorocarps, bases with halos and a trace of the early aggregation remains. Sori refruits, collapse soon or slides down and hang together; a frequent stoloniferous habit. (**E**), Round base (above, left); shortly digitate base with abundant mucilage sheath, a basal ring, or halo (above, right); young base terminating in a single larger cell, the slime is densely abundant and the sheath prolonged until the elevated ring or halo (below). (**F**), Piliform (left) and simple narrowed one-celled tip (right). (**G**), Short regular spores with polar to subpolar compound granules, some with halos. (**H**), Myxamoeba. Bars: (**A**,**B**) = 200 µm; (**C**,**D**) = 0.5 mm; (**E**) = 25 µm; (**F**) = 10 µm; (**G**) = 5 µm; (**H**) = 10 µm.

**Figure 7 jof-08-00593-f007:**
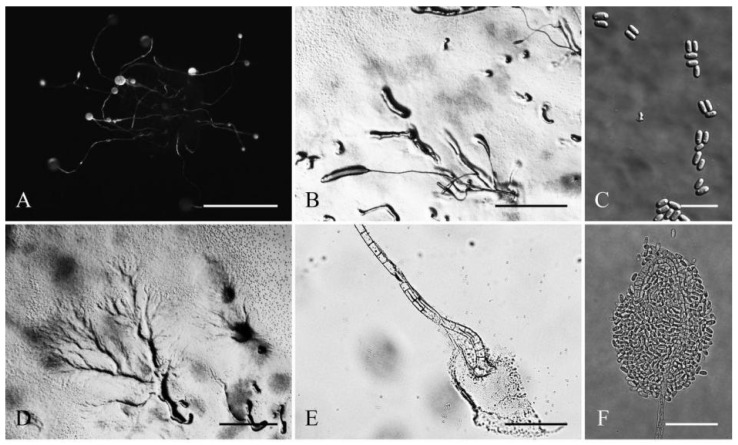
Morphological features of *Cavenderia ungulata* (TH18B). (**A**), Sorocarps. (**B**), Clustered pseudoplasmodia. (**C**), Spores. (**D**), Aggregations. (**E**), Sorophore base. (**F**), Sorophore tip. Bars: (**A**,**B**) = 1 mm; (**C**) = 20 μm; (**D**) = 1 mm; (**E**) = 60 μm; (**F**) = 40 μm.

**Figure 8 jof-08-00593-f008:**
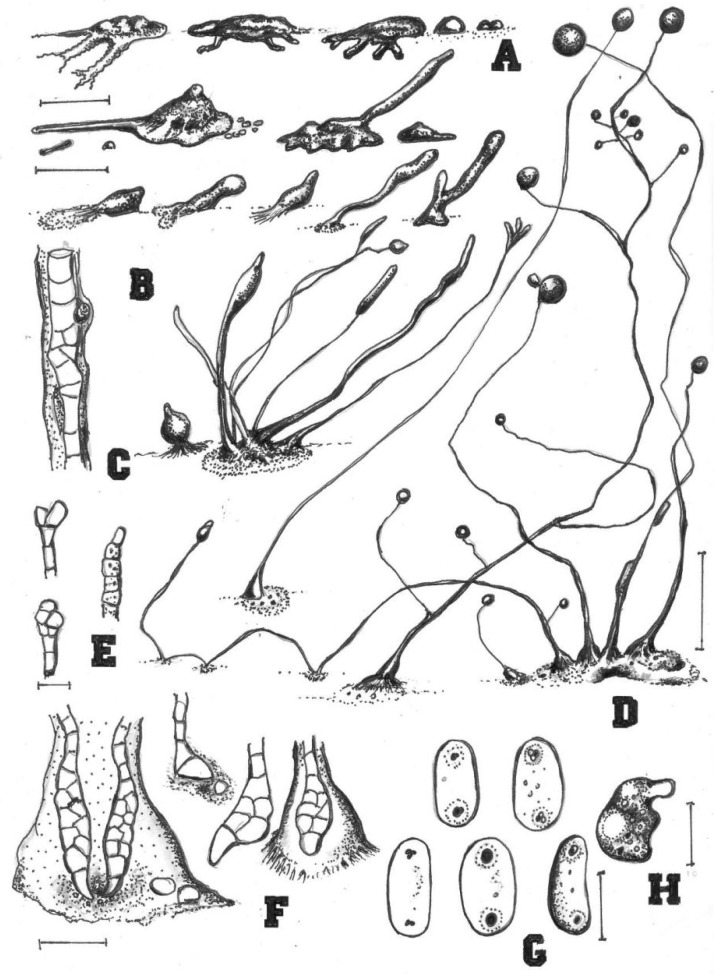
Morphological features of *Cavenderia ungulata* (TH18B). (**A**), Small to medium aggregations with short thick streams (left, center) and irregular small mounds (right). (**B**), An early sorogen rises up from the center, one elongated pseudoplasmodium stream remains (above, left); one early-late sorogen emerges from the side of the mound (above, right); from left to right: solitary migrating early sorogens (center); a cluster of late sorogens (below, right). (**C**), Irregular segment of a sorophore. (**D**), Solitary unbranched mature sorocarp with mucilage rest at the base (left); curved sorocarp with some branches (center); clustered sorocarps with ascending masses of small pseudoplasmodia, on the left margin there is a stoloniferous habit that progress to the left (right). (**E**), Two types of capitates tips (left); a flexuous simple tip (right). (**F**), Two tightly clustered clavate bases with terminal cells as cat claws, the darken matrix of slime is very dense and with large granules of yellow pigment (left); two solitary bases with enlarged terminal cells (center); a clavate regular base with its basal slime and sheath (right). (**G**), Slightly large elliptical spores with protruding PG (regular in shape) generally of the same size in both poles, also with vacuoles and halos. (**H**), Myxamoeba with many small-medium vacuoles. Bars: (**A**) = 300 µm; (**B**) = 200 µm; (**C**–**E**) = 15 µm; (**D**) = 0.5 mm; (**F**) = 20 µm; (**G**) = 6 µm; (**H**) = 10 µm.

**Figure 9 jof-08-00593-f009:**
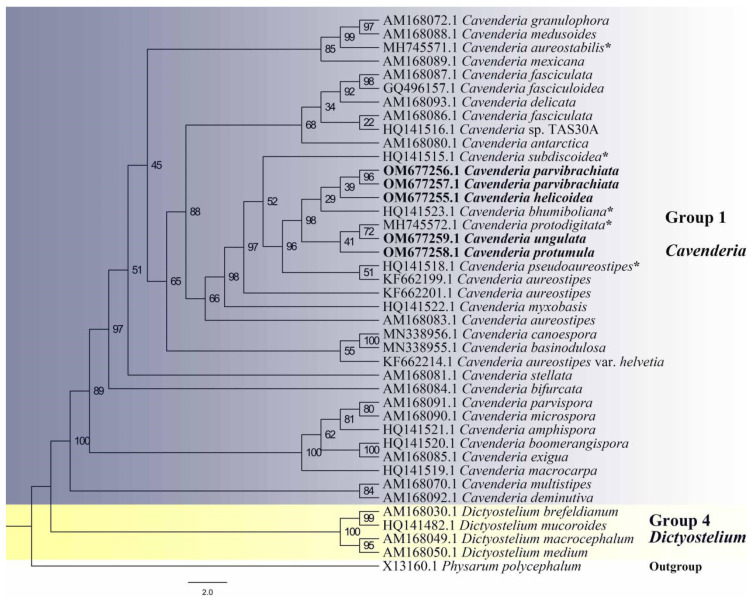
Phylogeny of four new species obtained in this study along with other species of *Cavenderia* based on SSU rRNA. Names in bold are the sequences obtained in this study, names with asterisks are the other five new species that were discovered and reported in this locality previously. The accession numbers are listed to the left of the species names. Complete strain information and GenBank accession numbers for all taxa in this tree can be observed in Table 1. The tree was derived using IQ-TREE with ultrafast bootstrap approximation (UFBoot) by TVMe+R5 model. The phylogeny is rooted according to Sheikh et al. [3].

**Table 1 jof-08-00593-t001:** NCBI GenBank accession information for SSU sequences of all isolates included in the phylogenetic analysis. New sequences are indicated in bold.

Taxon	Isolate No.	Accession No.
*Cavenderia amphispora*	BM9A	HQ141521.1
*C. antarctica*	NZ43B	AM168080.1
*C. aureostabilis*	ALP-2018a	MH745571.1
*C.aureostipes*	B15A	KF662199.1
*C. aureostipes*	YA6	AM168083.1
*C. aureostipes*	OH396	KF662201.1
*C. aureostipes* var. *helvetia*	HM592	KF662214.1
*C. basinodulosa*	ALP-2019a	MN338955.1
*C. bifurcata*	UK5	AM168084.1
*C. bhumiboliana*	THC11X	HQ141523.1
*C. boomerangispora*	K26B	HQ141520.1
*C. canoespora*	ALP-2019b	MN338956.1
*C. delicata*	TNS-C-226	AM168093.1
*C. deminutiva*	MexM19A	AM168092.1
*C. exigua*	TNS-C-199	AM168085.1
*C. fasciculata*	SH3	AM168087.1
*C. fasciculata*	SmokOW9A	AM168086.1
*C. fasciculoidea*	Cavender Puelo 1B	GQ496157.1
*C. granulophora*	CHII-4	AM168072.1
** *C. helicoidea* **	**TH19B**	**OM677255**
*C. macrocarpa*	MGE2	HQ141519.1
*C. medusoides*	OH592	AM168088.1
*C. mexicana*	MexTF4B1	AM168089.1
*C. microspora*	TNS-C-38	AM168090.1
*C. multistipes*	UK26b	AM168070.1
*C. myxobasis*	NT2A	HQ141522.1
** *C. parvibrachiata* **	**TH20C**	**OM677256**
** *C. parvibrachiata* **	**2019TH20C**	**OM677257**
*C. parvispora*	OS126	AM168091.1
*C. protodigitata*	ALP-2018b	MH745572.1
** *C. protumula* **	**TH20A**	**OM677258**
*C. pseudoaureostipes*	TH39A	HQ141518.1
*C.* sp.	TAS30A	HQ141516.1
*C. stellata*	SAB7B	AM168081.1
*C. subdiscoidea*	TH1A	HQ141515.1
** *C. ungulata* **	**TH18B**	**OM677259**
*Dictyostelium brefeldianum*	TNS-C-115	AM168030.1
*D. macrocephalum*	B33	AM168049.1
*D. medium*	TNS-C-205	AM168050.1
*D. mucoroides*	sweden 20	HQ141482.1

**Table 2 jof-08-00593-t002:** Summary data on a comparison of the new species with closely related species.

Species	Sorocarp	Sorophore	Sorophore Cell	Sorocarp Size (mm)	Base	Base Size	Tip	Tip Size	Branch	Sorous	Spore	Polar Granule	Aggregation	Yellow Pigmentation
*Cavenderia aureostabilis*	prone	slender to curved		L	Disk, round to clavate	S–L	unfinished capitate or small irregular cells	L		M	M–L	consolidated irregular	Radiate	Intense
*C. aureostipes*	erect	crowded		M	Round-irregular	M		M	>20	M	M	conspicuous	Polysphondylium violaceum type	Strong
*C. bhumiboliana*	prone	uneven, irregular	one tier	S	clavate, curved or not	L	Flexuous, piliform or round	L	1–4	S	L	prominent and large consolidated	Mounds	Fades
*C. helicoidea*	prone		one to several tiers	S	clavate to round	S–M	acuminate or piliform	S	a few	S–L	L	irregular consolidated	Radiate or taking the shape of irregular mounds	Fades
*C. parvibrachiata*	Erect to prostrate	slender	one or two tiers	S–L	round, clavate or curved hook-shaped	S–M	obtuse, capitate or clavate	S–M	Unbranched or irregular	S–L	M–L	pronounced consolidated polar-subpolar	Radiate	
*C. protodigitata*	Erect to prone	uneven	one tier	S	Clavate-digitate	S	piliform filaments or irregularly capitate	S	unbranched or secondary branched	S	S-M	two unequal medium to large consolidated, regular	Mounds	Fades
*C. protumula*	Erect		one to several tiers	S–L	clavate or round	M–L	obutuse, capitate	S–M	0–2	S–L	M–L	+	Radiate	
*C. ungulata*	erect to prone	very irregular, uneven	one or two tiers	S–M	claw-like clavate	S	variable, obtuse to acuminate or clavate	M	unbranched but sometimes with 1–5	S–L	S–L	large regular (2 µm)	Radiate	

Notes: Spores: S small (most common range: 4.5–5.5 × 2–3 μm), M median (most common range: 5.5–7.5 × 2.5–3.5 μm), L large (most common range: 6.5–9 × 3–5 μm) [3]. Sori: S small (most common range: 20–100 μm), M median (commonest range: 80–150 μm), L large (commonest range: 150–250 μm) [3]. Sorocarp Size: S small (commonest range: 0.2–3 mm), M median (commonest range: 3–5 mm), L large (most common range: 5–8 mm). Tip Size: S small (commonest range: 2–6 μm), M median (commonest range: 6.1–11 μm), L large (most common range: 15–30 μm). Bases Size: S small (commonest range: 8.4–20 μm), M median (commonest range: 20–30 μm), L large (most common range: 30–55 μm).

## Data Availability

The datasets generated for this study can be found in the GenBank and MycoBank.

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
