# Peer review of "Four New Species of Dictyostelids from Soil Systems in Northern Thailand"

_jof, 2022, doi:10.3390/jof8060593_

Round 1
Reviewer 1 Report
The study is well conceptualized and executed. Still I have some points to be clarified
- As the surveys and collections was conducted in 2008 but results are presented after 14 years.
- At some places the word sorocarp is used for the fruit structures of the collections, which is perfectly fine in case of slime mould but at a few places it is written as sporocarp, which needs to be corrected.
Reviewer 2 Report
In the article entitled “Four new species of dictyostelids from soil systems in northern Thailand”, the authors offer taxonomic descriptions of four new dictyostelid species isolated from Thailand, along with a molecular phylogenetic tree based on SSU rRNA. The paper is generally well written and taxonomic descriptions are concise and to the point. As four new species are described, this paper will be of interest to taxonomists, phylogeneticists, and evolutionary biologists interested in dictyostelids. I only have relatively minor concerns or suggestions.
(General issues)
1) The use of the term “group 1 species”.
The authors use the term “group 1 species” a few times without specifying what it means. I assume that this terminology is based on the SSU tree of Schaap et al 2006. If so, I think it is better to introduce it in the introduction, or at least cite the paper when you mention it to avoid confusion, because this paper follows the “new classification” by Sheikh et al 2018 in other occasions.
2) Discussion is relatively thin
Discussion is rather brief and does not touch on some of the issues which can be discussed. For instance,
- Based on the new species descriptions and molecular phylogenetics, what can we say about the features defining Cavenderia (is there any? or are they too variable?)?
- Is this locality particularly abundant with Cavenderia species compared to other regions (all the new species described so far from this place are Cavenderia, aren’t they?)?
- Can you offer any evidence that “organisms adapted to a cool environment” is “characterized by high levels of organic matter”? Also, what specifically are organic matters discussed here? The species descriptions did not give me an impression that these species are particularly rich in organic matters.
If you can address some of these issues, it will be interesting to non-taxonomist audience.
(Other issues regarding more specific points)
P.2 L 51: “Samples were processed by Landolt in his laboratory at Shepherd University.”
How? I think how it was done is more important than who did it. Is it the same as the procedure written in “2.2. Isolation and Cultivation”, or “processed” here referring to something different?
Also, if Landolt is the only one who processed the samples, isn’t it inconsistent with the “author contribution” section, which says “EMV, JCC, JCL and PL were responsible for the laboratory isolations and observations”?
P2. L55. “enough sterile distilled water added” → “enough sterile distilled water was added”
P.6. Table 2.“PG”
One of the columns in this table is labelled as “PG” but no explanation was given in the notes. I suppose that it signifies “polar granule”, but these abbreviations should be spelled out at first appearance.
P.7. Figure 9.
Figure 9 → Figure 1: I think the tree was moved during the editing process, but figures should be numbered in order of appearance.
“names in red are the sequences obtained in this study.” → names in bold?
It might be also useful to indicate on the tree other five new species that were found and reported in this locality previously (i.e., in Vadell et al 2018).
No support values are indicated on the nodes grouping C. ungulata, C. bhumiboliana, and C. Prodigitata.
P.16. L344.“This species adapts well to different conditions”.
What does it mean more specifically? Did you try various conditions (temperature, media, bacteria, etc) to grow and develop them?
P.19. L373, “In general, DNA sequence data have supported morphology-based species determination of dictyostelids. In fact, DNA sequence information recently has led to an extensive revision of the higher levels of classification within the dictyostelids [13].”
These sentences were confusing to me at first read. I suppose that the point is "DNA sequence data did not change the species-level classification much, but it did change the higher-level classification a lot". Then, should not “In fact” be "in contrast", “however”, “nevertheless”, etc instead?
Reviewer 3 Report
Dear authors,
I am kindly asking you to consider my comments and suggested corrections inserted as sticky-notes to your submitted pdf.
Especially take care of:
- Formating - check font type and font sizes, respect the formatting of the journal; renumber the Figures (your first Fig. is Fig. 9)
- Unclear statements - in several places I found unclear parts (i.e. in Material and Methods)
- Phylogenetic tree lacks basic information.
- Citation of your material examined - I advise to check in some of recently published articles in Journal of Fungi how studied material should be properly cited. You are citing holotypes but it is not clear where the material is deposited and if the "ex" collection is an isotype. Consult an expert in nomenclature/taxonomy if needed. Types should be clearly stated.
- Drawings are very nice and my impression in that they provide more information on the species comparing to the description. I would advise to improve the descriptions where possible using the characters that can be observed on the drawings.
These are some of my comments and all the rest you will find in the ms.
